# CREAGENTIVE: AN AGENT WORKFLOW DRIVEN MULTI-CATEGORY CREATIVE GENERATION ENGINE

## ABSTRACT

We present CreAgentive, an agent workflow driven multi-category creative generation engine that addresses four key limitations of contemporary large language models in writing stories, drama and other categories of creatives: restricted genre diversity, insufficient output length, weak narrative coherence, and inability to enforce complex structural constructs. At its core, CreAgentive employs a Story Prototype, which is a genre-agnostic, knowledge graph-based narrative representation that decouples story logic from stylistic realization by encoding characters, events, and environments as semantic triples. CreAgentive engages a three-stage agent workflow that comprises: an Initialization Stage that constructs a user-specified narrative skeleton; a Generation Stage in which long- and short-term objectives guide multi-agent dialogues to instantiate the Story Prototype; a Writing Stage that leverages this prototype to produce multi-genre text with advanced structures such as retrospection and foreshadowing. This architecture reduces storage redundancy and overcomes the typical bottlenecks of long-form generation. In extensive experiments, CreAgentive generates thousands of chapters with stable quality and low cost (less than \$1 per 100 chapters) using a general-purpose backbone model. To evaluate performance, we define a two-dimensional framework with 10 narrative indicators measuring both quality and length. Results show that CreAgentive consistently outperforms strong baselines and achieves robust performance across diverse genres, approaching the quality of human-authored novels.

## 1 INTRODUCTION

In recent years, the rapid advancement of large language models (LLMs) has reshaped the landscape of natural language generation (NLG) tasks(Tian et al., 2024). From poetry composition(Wang et al., 2025b), fiction writing(Huot et al., 2025), to research report generation(Xiong et al., 2025), LLMs have demonstrated remarkable capabilities in short-form creation (on the scale of a few thousand words). However, extending the application of LLMs to long-form narratives, such as serialized novels or multi-act screenplays, remains fundamentally challenging. As shown in Table 1, real-world creative writing tasks demand not only extensive length, but also diverse styles and complex narrative structures, representing critical bottlenecks for existing approaches. This gap stems from both the technical limitations inherent in the models(Liu et al., 2023) and the inherent complexity of the art of storytelling(Chakrabarty et al., 2024a), which together constrain the scale and depth of automated creative work.

Table 1: Comparison of narrative genres, typical word counts, and representative works.

| Genre | Typical Word Count | Representative Work |
|---|---|---|
| Web Novel | 2M–8M+ | *Worm* |
| Murder Mystery | 50k–100k | *Betrayal at House on the Hill* |
| Light Novel | 50k–100k per volume | *Mushoku Tensei* |
| Podcast Drama | 80k–150k per season | *Serial* |
| Short Drama Script | 60k–120k total | *Emma Approved* |
| Game Script | 100k–300k(main story) | *Genshin Impact* |

Current research indicates that existing LLM-based automated creative writing methods face four core limitations in long-form creative writing:

1. **Lack of Genre Diversity**: Most current systems are optimized for specific genres, making it difficult to effectively transfer the same story content across different genres (e.g., novels, screenplays, and poetry). This severely limits the ability of LLMs to generate diverse texts (Truong et al., 2025).

2. **Limited Long-Range Consistency**: The models struggle to manage long-range contextual information, which often leads to "hallucinations" in long-form generation. This can result in contradictory character behavior, fragmented plots, or inconsistent world building (Li et al., 2025).

3. **Output Length Constraints**: Due to the limitations of context windows, existing methods cannot generate a complete long-form text in a single pass. Repeated calls are not only inefficient, but can also weaken overall coherence (Mao et al., 2025).

4. **Lack of Complex Narrative Structures**: Most current models rely on linear plot progression, making it difficult to implement advanced narrative techniques such as nonlinear storytelling, multiple foreshadowing events, or chapter-to-chapter flashbacks and nested plots (Yu et al., 2025).

To systematically address these challenges, we propose ***CreAgentive: An Agent Workflow Driven Multi-Category Creative Generation Engine***. The core idea of CreAgentive is to decouple narrative logic from text generation, thereby supporting the creation of long-form, multi-genre, and complex narratives. To achieve this, we introduce the novel concept of Story Prototype, which uses a multi-version character plot dual knowledge graph to store and manage global narrative information. Building on this, CreAgentive is designed with a three-stage multi-agent workflow: the Initialization Stage sets the core theme, setting, and main character relationships based on user requirements; the Story Generation Stage plans the global narrative logic and plot development to generate the story prototype; and the Writing Stage transforms the story prototype into natural language text of the target genre. This framework design enables CreAgentive to operate independently of specific generative models or single methodologies. Instead, it can flexibly replace and integrate diverse generative components as needed, achieving both universality and scalability. Building upon this foundation, CreAgentive not only ensures coherence and consistency in long-form narratives but also holds future potential for constructing nonlinear narratives and other more complex narrative structures. Experimental results show that CreAgentive can efficiently generate long-form texts of millions of words or more at a relatively low generation cost, while supporting multiple genres. Its performance on key metrics such as generated length and narrative consistency significantly outperforms existing methods, fully validating the framework's practicality and adaptability for large-scale creative writing tasks.

The main contributions of this paper are as follows:

- We introduce the concept of Story Prototype for the first time, which decouples narrative logic from text generation through dual Character-Plot knowledge graphs. This approach provides a new paradigm for generating long-form, multi-genre, and complex narratives.

- We propose CreAgentive, a creative writing framework built on a three-stage multi-agent workflow. Supported by the Story Prototype, CreAgentive can effectively handle the generation of long-form, multi-genre, and complex narratives while maintaining high flexibility and scalability.

- We designed a systematic evaluation framework for long-form story generation. This framework combines human and automated evaluations to measure generation effectiveness across two dimensions, quality and length, using a total of 10 narrative indicators, thereby addressing a key gap in current research methodologies.

## 2 RELATED WORK

**Evolution of Story Generation Methods**. The field of automated story generation has evolved from symbolic planning and early neural models to approaches driven by Large Language Models (LLMs). A central challenge it faces is maintaining long-text consistency (Alabdulkarim et al.,

2021), which refers to the need to ensure logical coherence, factual plausibility, and world consistency as the complexity of the generated text increases. Early research enhanced narrative controllability via planning frameworks (Yao et al., 2019; Fan et al., 2019), followed by the emergence of outline-and-revise mechanisms that achieved more fine-grained control over long-form narratives (Yang et al., 2022a;b; Alhussain & Azmi, 2021). With the advent of LLMs, researchers have proposed methods such as explicit length control (Park et al., 2024) and extending the context window (Bai et al., 2024). The emergence of LLMs has further advanced the field, demonstrating immense potential (Wei et al., 2022; Coetzee, 2023). Concurrently, knowledge enhancement has become a focal point, with researchers attempting to incorporate external structured knowledge to improve coherence and factual grounding (Wang et al., 2023). Typical methods include collaborative generation between LLMs and knowledge graphs (Li et al., 2025; Pan et al., 2025; Zhou et al., 2024). Nevertheless, the output of existing methods is typically limited to a few thousand words, failing to meet the demands for long-form content such as novels or screenplays, which can range from tens of thousands to millions of words. Moreover, these approaches often exhibit genre-specificity, typically focusing on single formats such as novels (Huang et al., 2024), screenplays (Pichlmair et al., 2024), or poetry (Wang et al., 2025b), thereby limiting their applicability in cross-genre generation.

**Multi-Agent System of Story Generation**. In recent years, Multi-Agent Systems (MAS) have seen rapid development in fields like recommender systems, robotics, and social simulation (Zhang et al., 2024; Wang et al., 2025a; Mandi et al., 2024; Piao et al., 2025), significantly enhancing system intelligence through task decomposition and interaction (Zhang et al., 2023). Within the domain of story generation, MAS has demonstrated considerable potential (Xu et al., 2025). Notable systems include Agents' Room, StoryWriter, and BookWorld (Huot et al., 2025; Xia et al., 2025; Ran et al., 2025), with applications extending to drama and scriptwriting (Mirowski et al., 2023). Related research has also explored role-playing-based generation (Shao et al., 2023; Wang et al., 2024) and human-computer collaboration (Yuan et al., 2022; Ippolito et al., 2022; Calderwood et al., 2020; Li et al., 2024; Chakrabarty et al., 2024b; Hwang et al., 2025). However, existing methods still face challenges in simultaneously ensuring the continuability of long texts and the coherence of extended narratives.

**Evaluation of Story Generation**. The evaluation of story generation remains a significant challenge. Early methods primarily relied on human scoring (Guan & Huang, 2020; Hashimoto et al., 2019), which later evolved to include automatic metrics such as BLEU, ROUGE, METEOR, and BERTScore (Papineni et al., 2002; Lin, 2004; Banerjee & Lavie, 2005; Zhang et al., 2019). However, these metrics struggle to effectively measure narrative logic and creativity (Chhun et al., 2022; Liu et al., 2024b; Bohnet et al., 2024). More recently, "LLM-as-a-judge" approaches have been widely adopted (Gu et al., 2024), showing promise in assessing fluency and coherence. Yet, they may exhibit bias or inconsistent judgments on creativity (Zhou et al., 2025), and a unified evaluation framework is still lacking.

## 3 CREAGENTIVE

To systematically address the challenges in long-form, multi-genre, and complex narrative creation, we propose CreAgentive, we propose CreAgentive, an agent workflow driven multi-category creative generation engine. It abstracts "story" into a text-independent Story Prototype that decouples narrative logic from text generation, enabling cross-genre transfer, long-range coherence, and complex structure modeling. Based on this, CreAgentive designs a three-stage, multi-agent workflow: "Initialization — Story Generation — Writing", as shown in Figure 1.

### 3.1 STORY PROTOTYPE

A fundamental challenge in long-form creation is maintaining coherence across extensive texts. Existing methods often rely on outlines or paragraph-level generation, but these approaches are too rigid to support cross-genre transfer, complex plot interweaving, or long-text consistency. To address this, we introduce the Story Prototype as the core of CreAgentive. The Story Prototype is a genre-agnostic narrative representation that encodes characters, events, and scenes into semantic triples, forming a structured knowledge graph for precise narrative management and retrieval. It employs a Dual-Knowledge-Graph Synergy Structure consisting of the Role Graph and Plot Graph, which together serve as the semantic backbone of the global narrative. This design enables the

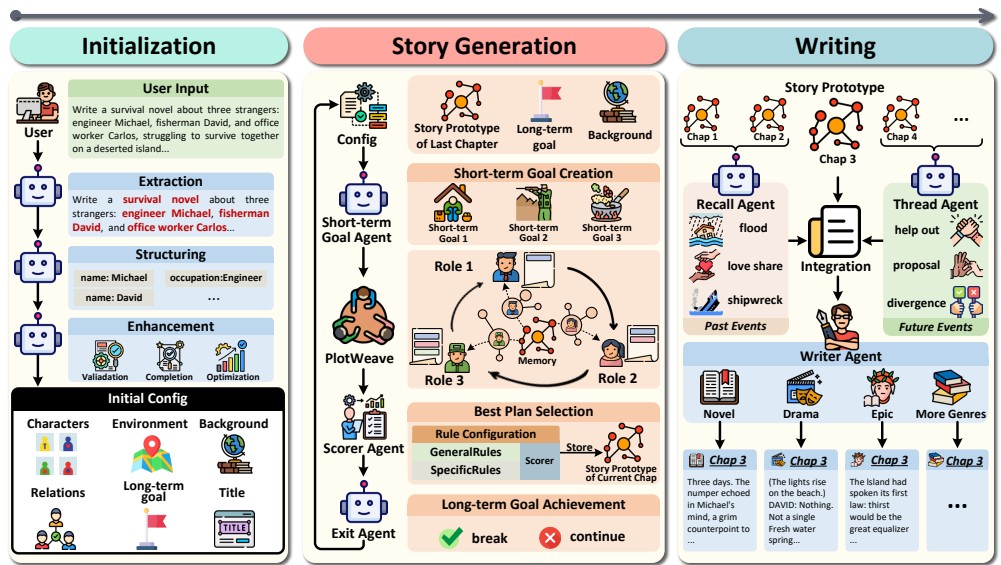

Figure 1: Overview of the CreAgentive. The system consists of three multi-agent workflows: Initialization, Story Generation, and Writing

joint management of character evolution and complex causal chains, which traditional outlines or single-graph methods struggle to achieve. The specific structure is shown in Figure 2.

**Role Graph**. The Role Graph is a character-centric dynamic relationship graph. Each node represents a character, containing static attributes (e.g., identity, gender, profession). Relationships between nodes include types like kinship and romantic relationships, and are equipped with properties such as strength, direction, and chapter labels, allowing relationships to evolve over time and capturing the subtle changes in character dynamics as the narrative progresses.

**Plot Graph**. The Plot Graph is a directed knowledge graph with events and scenes as basic units, tightly linked to roles. Roles are connected to events via "IN_EVENT" relationships, and events are connected to scenes via "OCCURRED_IN" relationships, collectively building a three-dimensional narrative structure. It is important to note that the Plot Graph records not only the basic information of each event and scene but also the consequences triggered by events and the specific emotional impact on each participant. This granular, individual-level impact tracking provides a basis for analyzing plot momentum and character changes from the overall story perspective.

This design delivers significant advantages:

- **Cross-Genre Versatility:** Since the Story Prototype stores abstract, genre-agnostic metadata, our system can seamlessly transform the same narrative prototype into multiple text formats, such as novels and screenplays.

- **Precise Narrative Retrieval:** The dual-graph joint index allows us to perform efficient queries to precisely retrieve specific character relationship evolutions or plot developments. This provides a solid foundation for subsequent agent decision-making and narrative enhancement.

- **Versioned Narrative Management:** We implement a versioned narrative through chapter-level prototype snapshots. This enables the system to trace back to historical versions and provides the possibility for creating complex non-linear narrative structures, such as flashbacks.

- **Decoupling and Coherence:** The Story Prototype decouples the narrative logic from the specific text implementation. It uses an intelligent version synchronization mechanism to ensure that while each chapter prototype evolves independently, the character attributes and global narrative remain coherent over a long span.

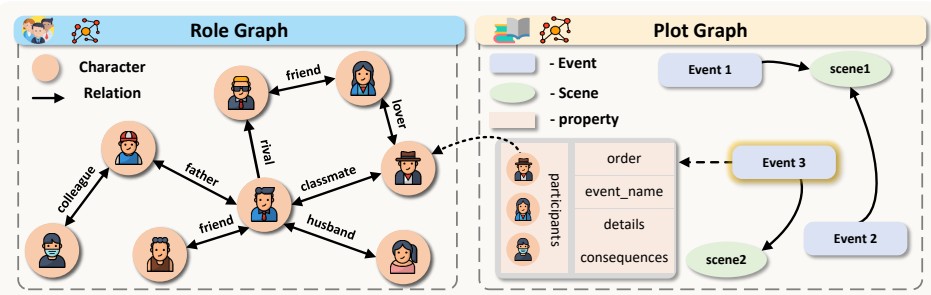

Figure 2: Illustration of the Story Prototype.

Unlike traditional outlines, the Story Prototype is not a text blueprint for writing but a genre-independent narrative abstraction layer. Outlines focus on chapter sequence and scene arrangement, belonging to the "how to write" category; the Story Prototype captures character motivations, causal chains, and background constraints, closer to the "story itself." This design enables CreAgentive to truly realize the creative paradigm of "story first, text later," providing a more solid foundation for long-form, multi-genre, and complex narratives.

### 3.2 THE INITIALIZATION WORKFLOW

This stage marks the starting point of the entire creative process. The Initialization Agent is responsible for converting the user's natural language input into a structured initial narrative configuration. This agent uses predefined templates to extract key information from the user's description, while also complementing and optimizing missing or incomplete information based on context and internal rules, thereby constructing a complete initial story setup (Initial Config). This mainly includes the characters and their relationships, the background setting, the long-term goal of the story, and the title with initial environment details.

After extraction, completion and optimization, the Initialization Agent writes this information into the Story Prototype, providing a complete and consistent global narrative framework for the subsequent Story Generation stage. This ensures the logical coherence and scalability of long-form text creation.

### 3.3 THE STORY GENERATION WORKFLOW

The Story Generation workflow in CreAgentive is designed to plan the global narrative logic and plot development, generating the Story Prototype. This workflow begins with the Short-term Goal Agent. Based on the Story Prototype, long-term goals, and overall background, this agent generates a set of short-term goals specific to the current chapter. Each goal represents a distinct path, designed to enrich the plot's development. Subsequently, the system assigns this set of goals to the Role Agents—automatically created based on the Story Prototype—to guide their subsequent actions.

In the PlotWeave, Role Agents (e.g., Role 1, 2, 3) collaborate to generate the plot around the short-term goal. Multiple Role Agents work together in a relay-style manner to weave the plot around this shared goal. This mechanism is fundamentally different from existing approaches like debate (Khan et al., 2024), competition (Cheng et al., 2024), or simple role-playing (Yu et al., 2025). Our design emphasizes that each agent, while maintaining its independent perspective, incrementally weaves the plot based on the contributions of the previous agent. This effectively avoids unnecessary conflicts and significantly improves generation efficiency. Throughout this process, all roles strictly adhere to the "Limited Cognition" principle, meaning each Role Agent can only access information in the Story Prototype directly related to its own character. This simulates the cognitive limitations of real individuals, preventing "omniscient perspectives" or logically inconsistent plots in the narrative.

To ensure the generated plot aligns with the Story Prototype and undergoes continuous optimization, we introduce the Scorer Agent. This agent quantitatively evaluates the candidate plots produced during the PlotWeave phase. Its scoring is based on predefined general rules (such as logical coherence, dramatic quality) and story-specific rules (such as consistency with character motivations), all of

which can be flexibly configured as needed. Ultimately, the highest-scoring proposal is adopted and written back into the Story Prototype for the current chapter.

Finally, the workflow is controlled by an Exit Agent. It checks against preset, verifiable conditions to determine whether the long-term goal has been achieved. If the condition is true, it terminates the story generation process; otherwise, the system proceeds to the next chapter's generation cycle. This mechanism forms a continuous, iterative, incremental generation loop, enabling it to effectively handle long-form narratives and fundamentally guarantee plot coherence.

### 3.4 THE WRITING WORKFLOW

The Writing Workflow aims to transform the Story Prototype into natural language text of a specific genre. The process can be divided into two stages. First, the system generates a detailed writing plan based on the current chapter's Story Prototype. To ensure textual depth and coherence, this stage involves the collaboration of the Recall Agent and the Thread Agent. The Recall Agent extracts relevant events and emotional memories from past chapters, providing depth and motivation for character actions. The Thread Agent analyzes preset key plots and foreshadowing from subsequent chapters, ensuring the current narration is tightly connected to future developments. This information is then integrated to form a complete Writing Plan. The complete Writing Plan is then executed by the Writer Agent. This agent, adhering to the user-specified genre and style, transforms the Story Prototype content into vivid, coherent natural language text, ensuring narrative logic and overall consistency. This multi-agent collaboration mechanism effectively addresses coherence challenges in long-form writing, offering a novel methodology for high-quality, multi-genre creative text generation.

## 4 EVALUATION

We propose HNES (Hierarchical Narrative Evaluation with State-Tracking), a comprehensive evaluation framework that assesses generated stories along two primary dimensions: content quality and narrative length. These dimensions are quantified through a quality score $S_q$ and a length score $S_l$ respectively. To provide a balanced overall assessment, we combine these scores into a composite metric—the Quality–Length Score (QLS)—defined as follows:

$$QLS = \frac{S_q + S_l}{2} \tag{1}$$

### 4.1 QUALITY DIMENSION EVALUATION

Drawing from existing research(Yang & Jin, 2024; Chakrabarty et al., 2024a), we evaluate story quality across seven narrative dimensions: Coherence (CH), Creativity (CR), Relevance (RE), Empathy (EM), Surprise (SU), Complexity (CX), and Immersion (IM)(see Appendix B for detailed definitions). Each dimension is rated on a scale of 1 to 10. The set of dimensions is denoted as $d \in D = \{CH, CR, RE, EM, SU, CX, IM\}$. As per prior studies, the weights for each dimension, $w_d$ are determined by the Analytic Hierarchy Process (AHP):

$$w_d = (w_{CH}, w_{CR}, w_{RE}, w_{EM}, w_{SU}, w_{CX}, w_{IM}) = (0.2, 0.2, 0.1, 0.15, 0.1, 0.1, 0.15)$$

Generated stories are scored using both automated and human-based methods. The scoring of generated stories incorporates both automated and human-based methods. For automated evaluation, we employ DeepSeek-R1 (Guo et al., 2025) as the base model, with the automated evaluation of HNES available in the Appendix C. For human evaluation, we engaged 5 literature enthusiasts, each with a strong literary background—including a TOEFL score exceeding 108 and having read more than 50 literary works—to independently score each dimension.

For each dimension $d$, we calculate an automated average score $\bar{A}_d$ and a human average score $\bar{H}_d$, then combine them to get the final score for that dimension $V_d$:

$$V_d = 0.5\bar{A}_d + 0.5\bar{H}_d \tag{2}$$

The final quality score, $S_q$ is the weighted sum of all dimension scores:

$$S_q = \sum_{d \in D} w_d V_d \tag{3}$$

## 4.2 Length Dimension Evaluation

The narrative length score, $S_l$ is based on the story's total word count $L_w$ and chapter count $L_c$. The scoring function is defined as:

$$S_l = \frac{1}{2} \left( \log \left( 1 + \frac{L_w}{1000} \right) + \min \left( 1, \frac{L_c}{C_{baseline}} \right) \right) \tag{4}$$

In this formula, the word count term uses a logarithmic form to moderately reward volume and prevent extremely long texts from dominating the score. The chapter term encourages a reasonable narrative structure. The final length score $S_l$ is an equal-weighted average of these two components, ensuring that both volume and structure contribute equally to the evaluation.

## 5 Experiment Setup

### 5.1 Baseline

We compare CreAgentive with representative open-source approaches covering three paradigms of story generation:

1. **Direct long-form generation models.** These methods generate stories end-to-end without structural control. We evaluate two representative approaches: *Direct*, which employs the base model with straightforward prompting, and *LongWriter-ChatGLM4-9B*, a pre-trained long-text generation model evaluated in its original form.

2. **Hierarchical generation methods.** These methods decompose story writing into multiple stages. We evaluate *DOC v2*[1] (document-level generation) and *Dramatron* (script generation from outlines).

3. **Multi-agent based generation methods.** These methods employ multiple agents to coordinate narrative planning and writing. We evaluate *Agents' Room*, which simulates collaborative story creation through agent interaction.

In addition, we include the real-world long-form web novel *Worm*[2] for serving as a human-authored reference.

### 5.2 Implementation

Framework-based models (Direct, DOC, Dramatron, Agents' Room) use DeepSeek-V3(Liu et al., 2024a) as the backbone, a large language model optimized for long-text generation and narrative understanding, while LongWriter-ChatGLM4-9B are used as pre-trained models. For Dramatron, which requires an outline for script generation, we first generate the outline using DeepSeek-V3 and then feed it into Dramatron for final story generation. Agents' Room is re-implemented using Autogen.CreAgentive is implemented based on the HAWK(Anonymous, 2025) framework using Autogen(Wu et al., 2024) and Neo4j[3] for structured memory and multi-agent coordination. All models are prompted with identical inputs, with the expected chapter count $C_{baseline}$ set to 10 for this experiment.

Table 2: Performance comparison of CreAgentive and baseline models.

| Model | Type | Quality Assessment | | | | | | | | Length | | | QLS |
|---|---|---|---|---|---|---|---|---|---|---|---|---|---|
| | | RE | CH | CR | EM | SU | CX | IM | $S_q$ | Words | Chap | $S_l$ | |
| *Direct long-form generation models* | | | | | | | | | | | | | |
| **Direct** | *Human* | 9.2 | 8.0 | 7.5 | 6.3 | 7.2 | 7.8 | 6.9 | 7.50 | 650 | 8 | 0.65 | 4.16 |
| | *Auto* | 8.3 | 7.9 | 8.7 | 7.7 | 7.4 | 7.1 | 7.2 | 7.84 | | | | |
| **LongWriter-chatglm4-9b** | *Human* | 8.1 | 8.6 | 7.3 | 8.4 | 6.6 | 7.0 | 7.7 | 7.76 | 2396 | 8 | 1.01 | 4.11 |
| | *Auto* | 7.0 | 6.3 | 8.1 | 6.3 | 5.7 | 5.8 | 6.5 | 6.65 | | | | |
| *Hierarchical generation methods* | | | | | | | | | | | | | |
| **DOC v2** | *Human* | 7.3 | 7.5 | 6.8 | 7.7 | 8.3 | 7.6 | 7.0 | 7.38 | 7391 | 4 | 1.26 | 3.92 |
| | *Auto* | 4.5 | 4.2 | 8.2 | 5.5 | 5.3 | 5.0 | 6.5 | 5.76 | | | | |
| **Dramatron** | *Human* | 8.2 | 7.1 | 7.1 | 7.0 | 7.5 | 8.2 | 7.2 | 7.36 | 653 | 8 | 0.65 | 3.82 |
| | *Auto* | 7.2 | 5.2 | 8.2 | 6.1 | 5.5 | 5.5 | 8.0 | 6.61 | | | | |
| *Multi-agent based generation methods* | | | | | | | | | | | | | |
| **Agents' Room** | *Human* | 7.5 | 9.1 | 8.5 | 7.3 | 7.6 | 8.5 | 7.3 | 8.07 | 3614 | 5 | 1.01 | 4.46 |
| | *Auto* | 8.0 | 7.2 | 8.6 | 7.7 | 6.8 | 6.8 | 8.5 | 7.75 | | | | |
| **CreAgentive(ours)** | *Human* | 8.7 | 8.5 | 8.8 | 7.8 | 8.0 | 8.7 | 7.4 | 8.28 | 4337 | 2770 | 1.34 | 4.78 |
| | *Auto* | 8.9 | 8.0 | 7.3 | 7.9 | 8.9 | 8.4 | 8.7 | 8.17 | | | | |
| *Human Writing* | | | | | | | | | | | | | |
| **Worm** | *Human* | 9.0 | 8.7 | 8.8 | 8.2 | 8.5 | 8.5 | 9.2 | 8.71 | 5158 | 105 | 1.41 | 4.96 |
| | *Auto* | 8.5 | 8.2 | 8.1 | 8.1 | 8.6 | 8.5 | 8.9 | 8.37 | | | | |

# 6 RESULTS

Our study comprises two complementary experimental components:

1. **Free-generation experiment**: All models—including CreAgentive and five baseline approaches spanning two narrative genres (novels and scripts)—receive identical user prompts and autonomously generate complete stories. Their overall generation quality is compared in Table 2;

2. **Per-chapter quality tracking experiment**: We dynamically evaluate narrative quality throughout the long-form generation process, continuously monitoring scores across seven core dimensions—Relevance, Coherence, Creativity, Empathy, Surprise, Complexity, and Immersion. The evolution of the aggregate Story Quality (SQ) score with increasing chapter count is visualized in Figure 3, and full dimension-wise results are provided in Appendix E.

**CreAgentive demonstrates outstanding performance in both human and automated evaluation.** As shown in Table 2, our framework achieves the highest quality scores from both assessment methods, with human evaluation yielding $S_q = 8.28$ and automated evaluation reaching $S_q = 8.17$. The strong alignment between these scores reflects remarkable consistency across evaluation methodologies. Notably, CreAgentive excels in key narrative dimensions, particularly creativity (CR: 8.8 human / 7.3 auto) and complexity (CX: 8.7 human / 8.4 auto), outperforming all baseline approaches. This convergence between human and automated assessments not only validates our evaluation framework but also confirms CreAgentive's exceptional narrative generation capabilities (see Appendix D for robustness across different base models as Judge). Moreover, compared with the human-authored novel *Worm*, CreAgentive's generation quality is already highly comparable and even surpasses it on certain indicators. This suggests that CreAgentive's narrative ability is approaching human level, marking a significant step toward automated long-form creative writing.

---

[1]https://github.com/facebookresearch/doc-storygen-v2

[2]https://parahumans.wordpress.com

[3]https://neo4j.com/

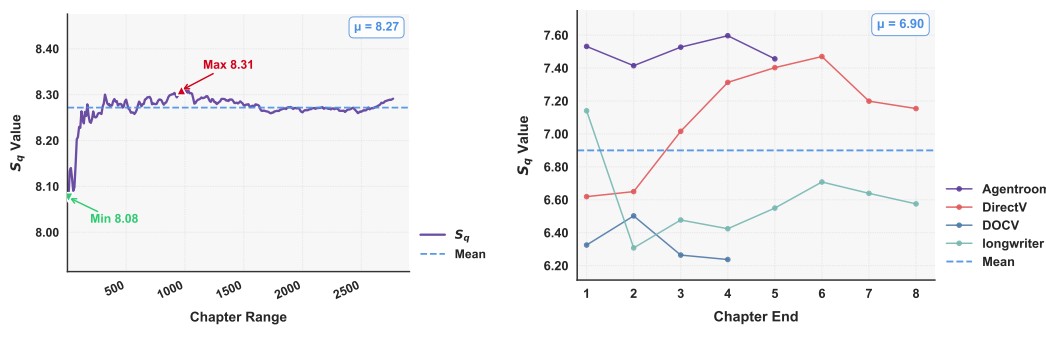

(a) CreAgentive across varying text lengths   (b) Baseline models across varying text lengths

Figure 3: Overall quality score $S_q$ across varying text lengths for CreAgentive and baseline models.

**CreAgentive maintains exceptional stability across varying narrative scales.** As shown in Figure 3a, the overall quality score $S_q$, remains consistently high even as the total number of generated chapters exceeds 2,500. The score fluctuates only minorly around a mean of $\mu = 8.27$, a remarkable consistency that demonstrates the framework's robustness in long-form generation. Further evidence in Appendix E reveals that all seven quality metrics—coherence, creativity, relevance, empathy, surprise, complexity, and immersion—maintain steady performance levels throughout the expansion process. No dimension exhibits abnormal fluctuations or declining trends, confirming the system's stable performance across scales. In contrast, as shown in Figure 3b, all baseline models generate significantly fewer chapters (at most 8) and exhibit substantially greater volatility in quality scores. Their average quality ($\mu = 6.90$) is notably lower than that of CreAgentive, and the scores fluctuate more widely across chapters, reflecting difficulties in maintaining narrative coherence and consistency over even short spans. This stark contrast underscores the limitations of existing approaches in long-form creative generation.

**Ablation Studies.** We conducted ablations on three components: short-term goals, PlotWeave, and Recall/Thread agents. As shown in Table F.1, removing any component lowered the overall quality score ($S_q$). In particular, excluding short-term goals reduced *Creativity* and *Surprise*, removing PlotWeave impaired *Coherence* and *Complexity*, and disabling Recall/Thread caused the largest drop, especially in *Coherence* and *Empathy*. These results demonstrate that all components contribute to maintaining coherent, high-quality long-form generation.

CreAgentive demonstrates notable efficiency in both time and cost. Additional experiments on base model preferences and cost efficiency (Appendix G, H) confirm its practicality: generation remains scalable and high-quality at low cost and moderate time. Furthermore, different base models yield consistent evaluation patterns, with DeepSeek-R1 closely aligned with other mainstream models, indicating no significant bias.

## 7 CONCLUSION

In this work, we introduced CreAgentive, an agent workflow-driven multi-category creative generation engine. At its core lies the Story Prototype, a genre-agnostic dual-knowledge-graph representation that decouples narrative logic from text realization. Through a structured three-stage agent workflow, CreAgentive guides narrative development from initialization and story generation to writing, ensuring consistency and coherence across long-form content. Extensive experiments demonstrate that CreAgentive significantly outperforms existing approaches in both quality and scalability, generating thousands of chapters of high-quality long-form content at minimal cost. Looking forward, we plan to extend CreAgentive to support more complex narrative structures such as interactive fiction and branching plots. We will also explore finer-grained control mechanisms for stylistic variation and emotional tone, and investigate its application in real-time collaborative human-AI writing scenarios.

## 8 ETHICS STATEMENT

Our research presents CreAgentive, a multi-agent system for automated narrative generation. While this technology enhances creative assistance, we acknowledge potential risks including content authenticity and misuse concerns. We encourage responsible deployment with proper attribution and transparency safeguards. Our goal is to augment human creativity, not replace it, and we emphasize the importance of ethical guidelines for positive societal impact.

## 9 REPRODUCIBILITY STATEMENT

To facilitate reproducibility, we provide source code at: `https://anonymous.4open.science/r/CreAgentive-761D`. Complete experimental implementation details are reported in Sections 5.2, with evaluation methodology detailed in Section 4 and Appendix C. These resources are provided to enable independent verification and extension of our work by the research community.

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

## A   THE USE OF LARGE LANGUAGE MODELS

Large language models (LLMs) were employed solely as an assistive tool for language refinement. Their usage was limited to improving grammar, readability, and stylistic clarity of the manuscript. No parts of the research process, including problem formulation, methodological design, experimental implementation, or result interpretation, involved the use of LLMs. All scientific content and conclusions are entirely the responsibility of the authors.

## B   QUALITY METRICS

Table B.1: Quality Metrics

| Metric | Explanation |
|---|---|
| *Story Core & Structure* | |
| Relevance(RE) | Assesses how well the story fits the initial prompt or theme. Does it stay focused and not stray from the topic? |
| Coherence(CO) | Evaluates the logical flow and consistency of the plot. Are the events smooth, without contradictions or breaks in the narrative, both within and between chapters? |
| Complexity(CX) | Evaluates the richness and depth of the story's structure. Are the plot and character relationships intricate and well-interwoven? |
| *Reader Experience & Emotion* | |
| Empathy (EM) | Measures the story's ability to evoke emotional connection with the reader. Are the characters believable and their struggles relatable? |
| Immersion (IM) | Assesses the level of detail and realism in the setting. Does the world-building pull the reader in and make them feel a part of the story? |
| *Creativity & Uniqueness* | |
| Surprise (SU) | Looks for unexpected plot twists or clever setups that go against the reader's expectations. Does the story have moments that genuinely surprise? |
| Creativity (CR) | Determines the originality of the story. Does it avoid common clichés and repetitive content, showcasing a unique concept? |

## C   AUTOMATED EVALUATION OF HNES

Conventional automated evaluation methods encounter substantial challenges when applied to long-form creative texts, primarily due to their high computational cost, limited scalability, and lack of interpretability. To overcome these limitations, we introduce the automated evaluation of HNES, which delivers fine-grained and interpretable assessments of long narratives in a cost-efficient and scalable manner.

### AGENT COLLABORATION

Automated evaluation integrates local evaluation and global evaluation through the collaboration of two specialized agents:

- **Chapter Analysis Agent (CAA):** Performs fine-grained evaluation of individual chapters and extracts their essential content.
- **Global Evaluation Agent (GEA):** Conducts holistic assessment of the narrative by leveraging the chapter summaries produced by CAA.

HIERARCHICAL EVALUATION MECHANISM

To enhance the accuracy of global evaluation, we introduces the notions of *Interval* and *Interval_Info*:

- **Interval:** denotes the number of chapters jointly assessed by GEA in a single batch.
- **Interval_Info:** refers to the cumulative narrative summary preserved after each global evaluation, which is subsequently utilized as contextual background for future assessments.

This hierarchical design ensures inherent scalability and efficiency, as GEA operates on refined summaries rather than the entirety of the raw text.

Periodically, when the interval condition is met, the GEA undertakes evaluation at the narrative's macro level (GEA.*run()*). It synthesizes multiple chapter summaries generated by CAA, thereby circumventing the computational burden of repeatedly analyzing raw text. GEA identifies cross-chapter thematic developments, character development, and narrative threads, producing a comprehensive account of story quality.

FRAMEWORK LOGIC AND KEY FUNCTIONS

The operational logic of the automated evaluation framework is formally detailed in Algorithm1 . The key functions and variables of this algorithm are defined as follows:

- INIT(`start_idx`, `chap_dir`): This function performs the initial setup of the workflow. It prepares the CAA and GEA agents and initializes the central state object, which serves as the memory for the entire process, holding all scores, extracted features, and contextual summaries.
- LOAD(`chap_dir`, `start_idx`, `end_idx`): This function is responsible for loading the dataset. It reads all chapter files from the specified directory, sorts them numerically, and selects the required range of chapters to be processed.
- UPDATE(`state`, `result`): This function is central to the state-tracking mechanism. After either agent runs, this function is called to integrate the new *local_result* or *global_result* into the central state object. This is how narrative context is built and the *Interval_Info* is progressively updated.
- REPORT(`state`, `chap_dir`): This function takes the final, fully populated state object and generates the framework's persistent output: a set of quantitative evaluation scores derived from the entire process.
- `state`: This is the core data structure that embodies the "State-Tracking" in HNES. It is a persistent object passed through the workflow that accumulates all results and context, including the *Interval_Info* and objective world conditions mentioned previously.

PARAMETERIZATION AND DYNAMIC ADJUSTMENT

**Note:** Our automated evaluation framework incorporates two key parameters with dynamic default values to adapt to different stages of narrative development:

- **Weight Allocation:** When calculating the final composite score, the weights assigned to local and global evaluation scores are dynamically adjusted. For the initial 10% of the story's chapters, this weight ratio is set to 4:1 (local:global) to emphasize the foundational plot construction. Subsequently, the ratio is adjusted to 1:1, giving equal importance to the macro-narrative structure and local details.
- **Evaluation Interval:** The number of chapters assessed by the Global Evaluation Agent (GEA) in a single batch defaults to 10 chapters. This means the GEA conducts a comprehensive evaluation of the story's macro-level progress at 10-chapter intervals.

Thus, by combining CAA and GEA, automated evaluation of HNES implements a dual-level evaluation framework that provides precise, interpretable, and actionable feedback for long-form creative writing.

---

**Algorithm 1** HNES Framework

---

**Require:** $chap\_dir$, $interval$, $start\_idx$, $end\_idx$

$\triangleright$ *1. Setup Phase*

1: $state, CAA, GEA \leftarrow \text{INIT}(start\_idx, chap\_dir)$ $\quad\quad\quad\quad$ $\triangleright$ Initialize state and agents
2: $chaps \leftarrow \text{LOAD}(chap\_dir, start\_idx, end\_idx)$ $\quad\quad\quad\quad$ $\triangleright$ Load chapters to process

$\triangleright$ *2. Processing Loop*

3: **for** each $chapter$ in $chaps$ **do**
4: $\quad$ $local\_result \leftarrow \text{CAA.run}(state, chapter)$ $\quad\quad\quad\quad$ $\triangleright$ Always run local analysis
5: $\quad$ $\text{UPDATE}(state, local\_result)$
6: $\quad$ **if** $interval$ is met **then**
7: $\quad\quad$ $global\_result \leftarrow \text{GEA.run}(state)$ $\quad\quad\quad\quad$ $\triangleright$ Periodically run global evaluation
8: $\quad\quad$ $\text{UPDATE}(state, global\_result)$
9: $\quad$ **end if**
10: **end for**

$\triangleright$ *3. Reporting Phase*

11: $\text{REPORT}(state, chap\_dir)$ $\quad\quad\quad\quad$ $\triangleright$ Calculate final scores and save files
12: **return** final scores

---

> **[Chapter Analysis Agent] Prompt**
>
> [Role]
> You are a professional literary analysis and story structure evaluation expert, skilled in extracting core plot elements from the text and providing precise scoring based on established literary criteria.
>
> [Your task]:
> Based on the provided [surface features of previous chapters] and [full content of this chapter], first extract the surface features of this chapter, and then give partial scores for the seven literary indicators focusing on the content of this chapter.
>
> [Definition of Surface Features]:
> 1.Unembellished plot summary:
> Describe the main characters, locations, events, and event outcomes of this chapter in concise, objective language.
> 2.Objective conditions at the end of the chapter:
> Includes but is not limited to changes in material quantities, character relationship status, geographical location shifts, and task progress.
>
> [Definition of the Seven Literary Indicators] (0–10 points each):
> 1.Relevance: Whether the story closely adheres to the given premise and thematic setting.
> 2.Coherence: Whether the plot in this chapter is logically consistent, flows naturally, and does not contradict previous chapters.
> 3.Empathy: Whether the characters are believable and can evoke emotional resonance in the reader.
> 4.Surprise: Whether it contains unexpected plot twists or clever setups.
> 5.Creativity: Whether the plot is original and avoids repetition.
> 6.Complexity: Whether the plot structure and character relationships are multilayered and contain narrative depth.
> 7.Immersion: The degree of detail in the environment and setting, and whether it can immerse the reader.
>
> [Strict Scoring Rules]:
> 1.Score Precision and Range:
> - The seven indicators allow two decimal places (e.g., 6.25).
> - Scores must accurately reflect the chapter's performance; any "comfort scoring" or

deliberate inflation is strictly prohibited.

2.Chapter Content as the Core, Previous Features as Supplement:
Scoring should be based mainly on the actual content of this chapter, not the overall story or earlier chapters. Previous chapters are used only to check logical consistency or relevance, not to boost scores.

3.Treatment of Plain or Ordinary Chapters:
For chapters lacking significant conflict, twists, emotional portrayal, or novel settings, strictly assign mid-to-low scores. Chapters with only minor highlights or small details must not exceed 8.00 in any indicator.

4.Handling of Surprises or Highlights:
High scores for Surprise, Creativity, and Complexity can only be given when the chapter contains clear and reasonable plot twists, original ideas, or emotional resonance. Minor changes, generic tropes, or common plot developments should not be mistaken as highlights.

5.Independent and Objective Scoring:
Each indicator must be scored independently; do not increase one score because another is high. All scores must be based on verifiable facts from the chapter, with no subjective bias.

6.Baseline Scoring:
- All indicators start at 6 points. If an indicator's performance is mediocre or has obvious flaws, the score should be below 6.
- A score above 9 indicates world-class mastery in that indicator, with no shortcomings in other aspects.

[Notes]:
1.The plot summary must be concise, objective, and free of embellishment.
2.Scores must be based on the chapter text and known plot context; do not fabricate content.
3.Do not add extra literary commentary; output only in the specified format.

[Output Format Requirement](you must follow this format strictly, don't add any extra explanation):

```
{
  "Surface Features": {
    "Plot Summary": "...",
    "Current Objective Conditions": "..."
  },
  "Partial Scores": {
    "Relevance": score,
    "Coherence": score,
    "Empathy": score,
    "Surprise": score,
    "Creativity": score,
    "Complexity": score,
    "Immersion": score
  }
}
```

[Global Evaluation Agent] Prompt

[Role] You are a professional literary work analysis and overall story quality evaluation expert, skilled in providing global scoring and structured summaries based on core elements from multi-chapter plots.

[Your task]:
Based on [surface features of all chapters], and considering the overall story development, provide a global score for the seven indicators and generate a structured story summary.

[Definition of the Seven Literary Indicators] (0–10 points each, allowing half points):
1.Relevance: Whether the whole book adheres closely to the given premise and thematic setting.
2.Coherence: The performance of the whole book in plot connection, character development, and logical consistency.
3.Empathy: Whether the overall story can make readers emotionally resonate with the characters.
4.Surprise: Whether the whole book contains unexpected plot twists or clever setups.
5.Creativity: Whether the whole book demonstrates originality and avoids overused tropes.
6.Complexity: The multilayered and intertwined nature of the story's plot structure and character relationships.
7.Immersion: Whether the book's overall world-building and setting are detailed enough to create an immersive experience.

[Scoring Standards]:
1.Score Precision and Limitations:
The seven indicators allow two decimal places (e.g., 6.25).
Do not artificially inflate scores; they must reflect the true quality of the work.
If an indicator is plain, ordinary, or lacks highlights, assign mid-to-low scores (usually in the 3–6 range).
Scores above 7.0 require solid content-based justification.

2.Use Chapter Surface Features Only:
- All scores must be strictly based on the provided chapter surface features.
- Do not assume or reference information not given in the summaries.
- The overall score must not be significantly increased because of a few standout chapters.

3.Indicator Independence:
Each indicator must be scored independently; do not raise one score because another is high. For example, if Surprise is low, it should not be raised because Immersion or Creativity is high.

4.Handling Highlights and Flaws:
Give high scores only for genuinely outstanding plot twists, original concepts, or complex relationships. Penalize for lack of highlights, flat plots. Minor changes, common tropes, or ordinary developments must not be mistaken as highlights.

5.Global Perspective Requirement:
Consider the work's overall thematic unity, narrative consistency, character development continuity, and structural completeness. Deduct points for plot holes, unreasonable character actions, or contradictions in the setting.

6.Baseline Scoring:
All indicators start at 6 points. If an indicator is mediocre or has obvious flaws, score it below 6. Scores above 9 indicate exceptional world-class mastery, with no shortcomings in other aspects.

[Notes]:
1.You must base the scoring strictly on the provided chapter surface features.
2.Consider thematic unity, narrative consistency, and structural completeness.
3.Do not output any extra explanation; output only in the specified format.
4.Do not use any markdown characters in your output; follow the exact format.

[Story Summary Content and Structure] (with strict constraints):
You must provide the story summary as a nested JSON object with the following three keys. The content for each key must adhere to the strict constraints described below:

1. Overall Synopsis:
- Constraint: Must be a single, concise sentence. Strictly no more than 40 words.
- Content: Summarize the novel's core background, the protagonist, and their fundamental motivations.

2. Main Characters Status Update:
- Constraint: Must list no more than the 3 most critical and currently active characters (protagonist and up to two others). The description for each character must be extremely brief. The entire string for this key should not exceed 100 words.
- Content: A string containing the list of these key characters. For each character, include their name, role, and a very brief summary of their current situation. Use "\n" and "\t" for formatting. Focus only on their immediate status and goals; do not include past events or resolved information.

3. Current Plot Status:
- Constraint: Must be a single, concise sentence. Strictly no more than 50 words.
- Content: Summarize the main plot's immediate state, highlighting the most direct crisis or cliffhanger at the end of the provided chapters. Do not describe past plot points.

[CRITICAL OUTPUT INSTRUCTIONS]:
- Your entire output MUST be a single, valid, parsable JSON object.
- The value for "Story Summary" MUST be a nested JSON object.
- You MUST strictly adhere to all word count and character count limits specified above. Your response will be rejected if it violates these constraints.
- Within the "Main Characters Status Update" string, all formatting MUST use escape characters ('\n' for newlines, '\t' for tabs).
- Do NOT output any extra explanation or markdown (like ```json) before or after the JSON object.

[Output Format Requirement](you must follow this format strictly, don't add any extra explanation):

```
{
  "Global Scores": {
    "Relevance": "score",
    "Coherence": "score",
    "Empathy": "score",
    "Surprise": "score",
    "Creativity": "score",
    "Complexity": "score",
    "Immersion": "score"
  },
  "Story Summary": {
    "Overall Synopsis": "...",
    "Main Characters Status Update": "...",
    "Current Plot Status": "..."
  }
}
```

## D  DIFFERENT MODEL PREFERENCES EXPERIMENTS

Table D.1: Comparison of writing quality across different base models under various generation frameworks. Each block corresponds to one narrative generation method (Direct prompting, Long-Writer, DOC v2, Dramatron, Agents' Room, and CreAgentive). Within each block, we report evaluation scores using different base models across seven narrative dimensions—Relevance (RE), Coherence (CH), Creativity (CR), Empathy (EM), Surprise (SU), Complexity (CX), and Immersion (IM)—along with the aggregated score $S_q$. This table highlights the sensitivity of different frameworks to the choice of base model.

| Model | Base Model | Quality Assessment | | | | | | | |
|---|---|---|---|---|---|---|---|---|---|
| | | RE | CH | CR | EM | SU | CX | IM | $S_q$ |
| **Direct** | DeepSeek-R1 | 8.3 | 7.9 | 8.7 | 7.7 | 7.4 | 7.1 | 7.2 | 7.84 |
| | DeepSeek-V3-0324 | 8.5 | 8.2 | 8.8 | 7.6 | 7.8 | 7.7 | 8.3 | 8.18 |
| | GPT5-mini | 7.7 | 6.3 | 8.3 | 6.8 | 6.2 | 6.4 | 6.3 | 6.92 |
| | Gmini2.5-Flash-Lite | 8.0 | 8.5 | 9.5 | 7.2 | 7.3 | 6.7 | 6.8 | 7.90 |
| | Qwen3-30B-A3B | 7.6 | 7.2 | 7.9 | 6.8 | 6.6 | 7.0 | 7.3 | 7.26 |
| **LongWriter-chatglm4-9b** | DeepSeek-R1 | 7.0 | 6.3 | 8.1 | 6.3 | 5.7 | 5.8 | 6.5 | 6.65 |
| | DeepSeek-V3-0324 | 7.8 | 7.1 | 7.9 | 7.2 | 6.9 | 6.5 | 6.8 | 7.22 |
| | GPT5-mini | 6.5 | 5.2 | 7.8 | 5.9 | 4.6 | 4.9 | 5.5 | 5.91 |
| | Gmini2.5-Flash-Lite | 7.9 | 6.2 | 8.7 | 6.7 | 4.9 | 5.5 | 6.3 | 6.76 |
| | Qwen3-30B-A3B | 7.3 | 6.6 | 7.0 | 6.6 | 6.0 | 6.3 | 7.2 | 6.75 |
| **DOC v2** | DeepSeek-R1 | 4.5 | 4.2 | 8.2 | 5.5 | 5.3 | 5.0 | 6.5 | 5.76 |
| | DeepSeek-V3-0324 | 7.4 | 7.0 | 7.9 | 6.8 | 6.7 | 6.6 | 7.7 | 7.22 |
| | GPT5-mini | 5.0 | 4.3 | 7.5 | 5.5 | 4.5 | 4.3 | 5.5 | 5.39 |
| | Gmini2.5-Flash-Lite | 6.9 | 6.3 | 8.1 | 5.7 | 5.8 | 5.5 | 6.2 | 6.48 |
| | Qwen3-30B-A3B | 6.6 | 6.0 | 6.6 | 5.7 | 5.7 | 5.5 | 6.4 | 6.12 |
| **Dramatron** | DeepSeek-R1 | 7.2 | 5.2 | 8.2 | 6.1 | 5.5 | 5.5 | 8.0 | 6.61 |
| | DeepSeek-V3-0324 | 7.2 | 6.6 | 7.5 | 6.5 | 6.3 | 6.3 | 7.8 | 6.94 |
| | GPT5-mini | 6.3 | 4.8 | 7.6 | 4.8 | 3.7 | 3.8 | 6.9 | 5.62 |
| | Gmini2.5-Flash-Lite | 7.8 | 6.5 | 8.1 | 5.9 | 5.4 | 4.8 | 7.1 | 6.67 |
| | Qwen3-30B-A3B | 7.0 | 6.4 | 7.0 | 6.3 | 6.0 | 6.0 | 7.1 | 6.59 |
| **Agents' Room** | DeepSeek-R1 | 8.0 | 7.2 | 8.6 | 7.7 | 6.8 | 6.8 | 8.5 | 7.75 |
| | DeepSeek-V3-0324 | 8.3 | 7.7 | 8.3 | 8.4 | 7.5 | 7.5 | 8.3 | 8.04 |
| | GPT5-mini | 7.4 | 6.6 | 7.6 | 6.6 | 5.8 | 6.3 | 7.0 | 6.83 |
| | Gmini2.5-Flash-Lite | 7.6 | 6.8 | 9.5 | 7.0 | 6.2 | 6.2 | 6.3 | 7.26 |
| | Qwen3-30B-A3B | 7.5 | 6.7 | 7.3 | 7.0 | 6.6 | 7.1 | 7.6 | 7.11 |
| **CreAgentive(ours)** | DeepSeek-R1 | 7.9 | 8.5 | 8.7 | 7.2 | 7.5 | 8.2 | 8.2 | 8.11 |
| | DeepSeek-V3-0324 | 8.7 | 8.6 | 8.7 | 7.4 | 7.9 | 8.8 | 8.3 | 8.35 |
| | GPT5-mini | 7.5 | 7.7 | 8.0 | 5.5 | 7.0 | 7.1 | 7.9 | 7.31 |
| | Gmini2.5-Flash-Lite | 8.1 | 8.0 | 9.0 | 6.4 | 7.2 | 7.6 | 7.2 | 7.73 |
| | Qwen3-30B-A3B | 7.2 | 7.2 | 7.0 | 6.3 | 6.8 | 6.7 | 7.3 | 6.95 |

# E    QUALITY ROBUST

Table E.1: Representative quality scores of CreAgentive across different chapter ranges. The table reports evaluation results on seven narrative quality dimensions—Relevance (RE), Coherence (CH), Creativity (CR), Empathy (EM), Surprise (SU), Complexity (CX), and Immersion (IM)—along with the aggregated score $S_q$. To illustrate the stability and robustness of the model's performance, we present selected chapters as representative checkpoints rather than exhaustively reporting every step.

| Chap | Quality Assessment | | | | | | | |
|---|---|---|---|---|---|---|---|---|
| | RE | CH | CR | EM | SU | CX | IM | $S_q$ |
| *10 - 50* | | | | | | | | |
| 10 | 8.6 | 8.0 | 8.3 | 8.2 | 7.1 | 7.7 | 8.1 | 8.7 |
| 20 | 8.8 | 7.9 | 8.4 | 7.3 | 7.6 | 8.3 | 8.7 | 8.1 |
| 30 | 8.8 | 8.1 | 8.3 | 7.3 | 7.5 | 8.4 | 8.7 | 8.1 |
| 40 | 8.7 | 7.9 | 8.3 | 7.4 | 7.7 | 8.3 | 8.7 | 8.1 |
| 50 | 8.7 | 7.9 | 8.3 | 7.4 | 7.5 | 8.2 | 8.6 | 8.1 |
| *100 - 550* | | | | | | | | |
| 100 | 8.8 | 7.9 | 8.7 | 7.3 | 7.9 | 8.4 | 8.6 | 8.2 |
| 150 | 8.8 | 8.0 | 8.8 | 7.2 | 8.1 | 8.5 | 8.6 | 8.3 |
| 200 | 8.8 | 8.0 | 8.7 | 7.3 | 8.0 | 8.4 | 8.6 | 8.2 |
| 250 | 8.8 | 7.9 | 8.7 | 7.4 | 8.1 | 8.5 | 8.6 | 8.3 |
| 300 | 8.8 | 8.0 | 8.7 | 7.4 | 8.1 | 8.5 | 8.6 | 8.3 |
| 350 | 8.8 | 8.0 | 8.7 | 7.4 | 8.1 | 8.5 | 8.6 | 8.3 |
| 400 | 8.8 | 8.0 | 8.8 | 7.4 | 8.1 | 8.5 | 8.6 | 8.3 |
| 450 | 8.8 | 8.0 | 8.7 | 7.4 | 8.0 | 8.5 | 8.6 | 8.3 |
| 500 | 8.8 | 8.0 | 8.7 | 7.5 | 8.0 | 8.5 | 8.6 | 8.3 |
| 550 | 8.8 | 8.0 | 8.7 | 7.4 | 8.0 | 8.4 | 8.6 | 8.3 |
| *600 - 1000* | | | | | | | | |
| 600 | 8.8 | 8.0 | 8.7 | 7.4 | 8.0 | 8.5 | 8.6 | 8.3 |
| 700 | 8.9 | 8.0 | 8.7 | 7.5 | 8.0 | 8.5 | 8.6 | 8.3 |
| 800 | 8.9 | 8.0 | 8.7 | 7.5 | 8.0 | 8.4 | 8.6 | 8.3 |
| 900 | 8.9 | 8.0 | 8.7 | 7.5 | 8.0 | 8.4 | 8.6 | 8.3 |
| 1000 | 8.9 | 8.0 | 8.8 | 7.5 | 8.0 | 8.4 | 8.7 | 8.3 |
| *1500 - 2700* | | | | | | | | |
| 1500 | 8.9 | 8.0 | 8.8 | 7.5 | 8.0 | 8.4 | 8.7 | 8.3 |
| 2000 | 8.9 | 8.0 | 8.9 | 7.3 | 7.9 | 8.3 | 8.7 | 8.3 |
| 2500 | 8.9 | 7.9 | 8.9 | 7.3 | 7.9 | 8.4 | 8.6 | 8.3 |
| 2700 | 8.9 | 8.0 | 8.9 | 7.3 | 7.9 | 8.4 | 8.6 | 8.3 |

# F    ABLATION STUDY

Table F.1: Performance comparison of CreAgentive and its ablation variants

| Model | Quality Assessment | | | | | | | |
|---|---|---|---|---|---|---|---|---|
| | RE | CH | CR | EM | SU | CX | IM | $S_q$ |
| CreAgentive | 9.02 | 8.62 | 8.65 | 7.21 | 8.69 | 8.84 | 8.59 | 8.48 |
| (-) Multiple short-term goals | 8.65 | 8.11 | 8.40 | 7.22 | 8.24 | 8.70 | 8.16 | 8.17 |
| (-) Plotweave | 9.00 | 8.08 | 8.42 | 7.16 | 8.42 | 8.39 | 8.31 | 8.20 |
| (-) Recall and Thread | 8.55 | 7.52 | 8.50 | 6.67 | 8.30 | 7.75 | 8.46 | 7.93 |

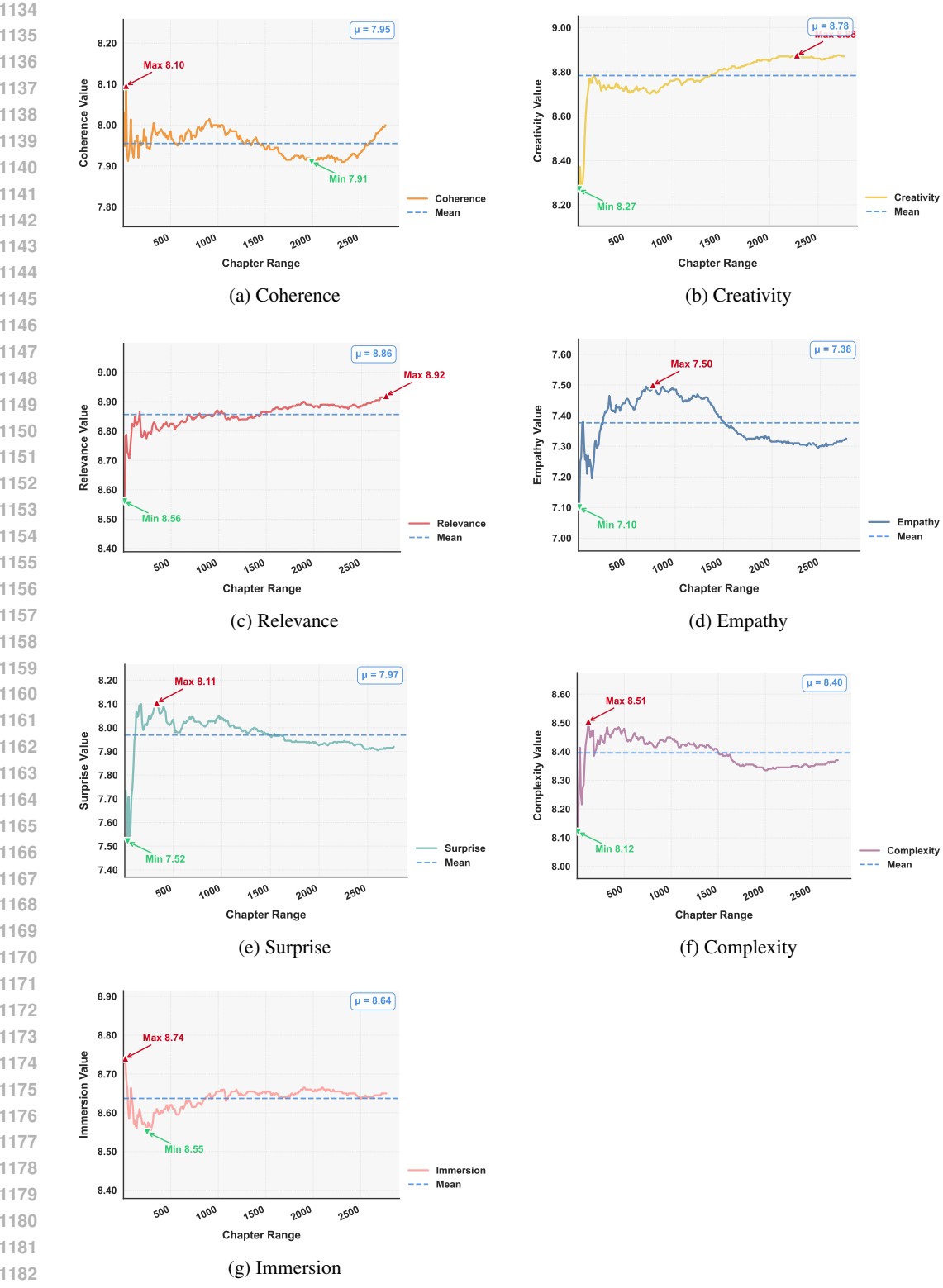

Figure E.1: Evaluation of CreAgentive across seven narrative quality dimensions as the number of generated chapters increases. Each subplot reports the average score evolution along a specific dimension: (a) Coherence, (b) Creativity, (c) Relevance, (d) Empathy, (e) Surprise, (f) Complexity, and (g) Immersion. The results indicate how CreAgentive maintains or improves performance in different aspects of story generation when scaling to longer narratives.

# G WRITING ABILITY OF DIFFERENT BASE MODEL

Table G.1: Different BaseModel's Writing Ability. The table shows the quality scores of stories generated by different base models given the same story prototype.

| Model | Quality Assessment | | | | | | | |
|---|---|---|---|---|---|---|---|---|
| | RE | CH | CR | EM | SU | CX | IM | $S_q$ |
| Deepseek-V3-0324 | 8.0 | 8.6 | 8.4 | 6.4 | 8.0 | 8.5 | 8.2 | 8.04 |
| GPT-5-mini | 8.1 | 6.7 | 8.7 | 7.9 | 6.9 | 8.3 | 8.7 | 7.90 |
| Gemini-2.5-flash-lite | 7.8 | 7.2 | 8.4 | 7.1 | 6.8 | 7.6 | 8.0 | 7.60 |
| Qwen3-30B-A3B | 7.4 | 7.8 | 8.2 | 6.6 | 8.1 | 8.2 | 8.0 | 7.76 |

# H COST

Table H.1: Cost and time efficiency of CreAgentive across different base models. The table reports the average monetary cost (USD/Chapter), time consumption (Minutes/Chapter), and output length (Words/Chapter) for both story generation and writing stages.

| Model | StorGen | | Writing | | All | | Ave Words/Chapter |
|---|---|---|---|---|---|---|---|
| | USD / Chap | Min / Chap | USD / Chap | Min / Chap | USD / Chap | Min / Chap | |
| Deepseek-V3-0324 | 0.1361 | 6.2 | 0.0178 | 0.9 | 0.1539 | 7.1 | 634 |
| GPT-5-mini | 1.2078 | 16.6 | 0.2450 | 2.7 | 1.4528 | 19.3 | 5270 |
| Gemini-2.5-flash-lite | 0.0131 | 1.6 | 0.0013 | 0.2 | 0.0152 | 1.8 | 1876 |
| Qwen3-30B-A3B | 0.0140 | 8.9 | 0.0004 | 1.5 | 0.0144 | 10.4 | 2506 |

