# OpenReview forum: "CreAgentive: An Agent Workflow Driven Multi-Category Creative Generation Engine"
_ICLR.cc/2026/Conference — Submitted to ICLR 2026_

### Official Review · Reviewer_tHf5 · 2025-10-23

**Soundness:** 3
**Presentation:** 2
**Contribution:** 2
**Rating:** 4
**Confidence:** 3

**Summary:**

This work presents CreAgentive, an agent workflow for generating stories. The workflow consists of three steps: Initialization, Story Generation, and Writing. Key components includes multiple role-playing agents collaboration, scorer agent, and exit agent. Their experiment results show CreAgentive outperform baselines on story quality and

**Strengths:**

1. The workflow consists of delicate operations to process user input and generate story, where lots of engineering effort seems to have been put.
2. Showing quality assessment across different narrative dimensions for fine-grained analysis.

**Weaknesses:**

1. **The workflow description is too generic and lacks clarity and details**. It would be better to elaborate on important steps like how the elements in the config are constructed, the data format for generated plots, how the short-term goals are generated etc.
2. **The workflow seems to be resource-intensive**. Almost each step is done with an agent, and the plotweave process requires multiple agents at the same time.
3. **Lack quality report on human evaluation**. How reliable of the human evaluation is unknown.
4. **Limited ablation study depth**: Given the number of design choices, a more detailed ablation would clarify each component’s contribution, but the current analysis is limited.
5. **Reliability of evaluation metrics**: The metrics $S_q$, $S_l$ seem to be customized-determined and could be biased.
6. **"Limited Cognition" principle in plotweave process**. This constraint may be stricter than typical human behavior. In practice, people maintain and update partial knowledge about others

**Questions:**

1. In section 3.3, could you share a concrete example or illustration of a PlotWeave? I’d also appreciate a bit more detail on the phrase “incrementally weaves the plot based on the contributions of the previous agent,” as I’m not fully grasping the mechanism.
2. For the PlotWeave process, do you set an upper bound on the number of role agents?
3. I think the "Limited Cognition" principle is too strict, because in realworld people actually know a lot information from others and update their knowledge of others through communication. Do you view this constraint as a potential limitation, and might a partial-observability/updateable memory variant be feasible?
4. In Eq4, could you clarify the meaning of 'C_baseline'?
5. In Table 2 (the result table), CreAgentive (ours) produces far more chapters than other baselines (over 2,000). How should we interpret this scale. Is it desirable?
6. Would you be able to share the user prompts used in the experiments?
7. Could you add human annotation quality analysis like inter-annotator agreement?

---

### Official Review · Reviewer_xMob · 2025-10-30

**Soundness:** 3
**Presentation:** 2
**Contribution:** 2
**Rating:** 2
**Confidence:** 4

**Summary:**

In this paper, the authors present CreAgentive, an agent for generating high-quality, long-form creative narratives across multiple genres. It aims to address common LLM limitations such as poor long-range coherence, short length, and weak structural control. The system’s novelty is the Story Prototype—a genre-agnostic dual knowledge graph that captures the narrative’s abstract logic (characters, relationships, and plot events) separately from the final text. CreAgentive uses a three-stage workflow: agents first initialize the story, then collaboratively plan the full plot by building the Story Prototype chapter by chapter, and finally a separate set of agents write the prose or script. By separating planning from writing, the system ensures logical consistency, supports complex structures like foreshadowing, and, as experiments show, can generate thousands of coherent chapters, outperforming all baselines and approaching human-level quality.

**Strengths:**

1. The system shows strong performance and importance through its results. While baseline models collapsed after about 8 chapters, CreAgentive generated over 2700 chapters with stable, high-quality scores. This proves its design effectively solves the long-range coherence problem for true, novel-length storytelling.

2. The Story Prototype is a dynamic, multi-version knowledge graph that serves as the single source of truth for all plot and character logic. This structure gives the system its long-range memory and ability to handle complex narrative structures.

**Weaknesses:**

1. The Section 3, especially Section 3.3 is very vague. I have no idea what is "Role Agents" and "Goal Agent". Only a GPT call with prompt? Also, the "Scorer Agent" seems to rely on undefined "rules," does it hide a potential human-in-the-loop cost?

2. No human-verification for metrics such as QLS. The 50/50 split between quality and length is an arbitrary assumption. How to prove this metrics align with human preference? Since authors claim HNES is a big novelty in the paper, the usefulness of these metrics are prerequisite for the experiment section.

3. Lack of evaluation that directly assesses the quality of the Story Prototype itself. The evaluation only measures the final generated text. I have no idea if the system's success comes from a high-quality prototype or from a powerful writer agent.

4. Comparison is fair? Such as for "Dramatron", it relies on outlines, so only 8 chapters mean the given outline is too short? This is a flaw in the experimental setup, not a failure of the baseline

**Questions:**

1. Dramatron, Agents' Room is designed for long story generation? Is the comparison fair? for Dramatron, it relies on outlines, so only 8 chapters mean the given outline is too short?

2. Did you use human to validate the LLM-judge? Seem you only use this as a part of final score. This is not a validation for the LLM-judge.

---

### Official Review · Reviewer_aDeD · 2025-11-01

**Soundness:** 2
**Presentation:** 1
**Contribution:** 1
**Rating:** 2
**Confidence:** 4

**Summary:**

The authors present CreAgentive, a multi‑agent writing system built around a “Story Prototype”, a knowledge graph representation that separates narrative logic from prose. They argue this abstraction enables cross‑genre transfer (novel, script), long‑range coherence, and advanced structures (retrospection/foreshadowing). A three‑stage workflow coordinates Short‑term Goal Agents, Role Agents with “Limited Cognition,” a Scorer, and Recall/Thread agents. For evaluation, they introduce HNES, which blends seven narrative quality indicators with a length reward into a composite QLS. They claim CreAgentive outperforms strong baselines, scales to thousands of chapters, approaches human‑authored novels, and does so at very low cost.

**Strengths:**

I think there are alot of good ideas with potential in this paper, but I have some major concerns about the claims and evaluation.

Let's start with some positives:
- The Story Prototype is a clear attempt to encode “story first, text later,” using a dual KG to track roles, events, consequences, and emotions. Figure 1 and Figure 2 communicate the intended control points and where memory lives. This architectural separation is a promising direction for long‑form generation.
- The PlotWeave procedure, Short‑term Goal Agent, Scorer, and Recall/Thread agents directly address drift, shallow causality, and weak foreshadowing.
- The paper emphasizes stability with scale via per‑chapter quality tracking out past 2,500 chapters
- Evaluation against a human baseline in 'Worm'

**Weaknesses:**

Weaknesses:

- Cost claim contradicts the appendix: The abstract is overclaiming a step too far here (less than 1 dollar per 100 chapters), if the cheapest case is GPT-5-mini with an estimated 0.0152 per chapter
-  QLS is defined earlier by the authors as $(S_q + S_l)/2$  (Eq. 1). if you plug this into Table 2 you get different values for several baselines and the human reference (e.i., DOC v2 “Human” $(7.38 + 1.26)/ 2 =  4.32$ expected vs. 3.92 printed)
-  Whats the word-chapter relationship here? Since there are no fractions, im assuming ‘words’ is the total length of the generation. But this means CreAgentive generated chapters on the length of roughly TWO words???
- If the above understanding is INCORRECT, why is CreAgentive allowed to generate so much longer than everything else? Isnt this a bit unfair? (See issues with length reward)
- Length reward (S_l) structurally benefits systems that can “just add chapters.” S_l averages a log‑word term with a chapter‑count term that saturates at L_c ≥ C_baseline (here 10). CreAgentive is run to thousands of chapters; baselines are stopped at ≤ 8 chapters, then compared on a metric that explicitly rewards length and chapterization.
- Human evaluation uses n=5 literature enthusiasts with no reported inter‑rater reliability
- The text points to Appendix D for “robustness across different base models as Judge,” but Appendix D actually varies writer base models across frameworks so judge‑sensitivity is not shown.
- My impression is that the work with the Story Prototype is extremely similar to the KG-Based story related work done by Mark Riedl’s lab (https://arxiv.org/abs/2404.13076, https://arxiv.org/abs/2112.08596). Could the authors clarify the key differences between their approach and, for example, the story plot skeleton?

**Questions:**

I've largely mixed my questions in with my criticisms in the 'weaknesses' section. I also invite the authors to clarify any misunderstandings I have. I will start with a lower score for now, but would like to emphasize an open mind towards upward adjustments.

---

### Meta-Review · Area_Chair_Z3Mv · 2026-01-08

**Summary:**

While the high level idea behind the paper is not contested by the reviewers, most of them had concerns about clarity, the evaluation, and corresponding claims being made. The automated evaluation numbers are not inconsistent (baselines for their QLS metrics get different numbers as pointed out by aDeD). The human evaluation being n=5 with no further metrics is also another big red flag. Further, it seems like the reviewers had trouble understanding the actual method employed due to a lack of clarity in the description.

**Reviewer Concerns:**

None of the concerns were addressed.

**Reviewer Scores:**

There was no rebuttal

---

### Decision · Program_Chairs · 2026-01-26

Reject